# Dry Addition of Recycled Waste Polyethylene in Asphalt Mixtures: A Laboratory Study

**DOI:** 10.3390/ma15144739

**Published:** 2022-07-06

**Authors:** Marco Pasetto, Andrea Baliello, Emiliano Pasquini, Lily Poulikakos

**Affiliations:** 1Department of Civil, Environmental and Architectural Engineering, University of Padova, Via Marzolo 9, 35131 Padova, Italy; andrea.baliello@unipd.it (A.B.); emiliano.pasquini@unipd.it (E.P.); 2EMPA, Swiss Federal Laboratory for Material Science and Technology, Überlandstrasse 129, 8600 Dübendorf, Switzerland; lily.poulikakos@empa.ch

**Keywords:** plastic waste, circular economy, pavement engineering, workability, linear visco-elastic characteristics, stiffness, fracture resistance, permanent deformation, moisture resistance

## Abstract

The circular use of resources (i.e., reuse and recycling of materials) aiming for zero waste is also gaining increasing attention in pavement engineering. In this regard, the possible use of waste plastics in asphalt materials is of strategic importance since a considerable amount of plastic waste from construction and demolition waste and municipal solid waste is generated every year. Given this background, this experimental study aimed to investigate the feasibility of recycling waste polyethylene (PE) into asphalt mixtures. For this purpose, the dry addition of plastic shreds was evaluated to overcome the drawbacks observed in a previous interlaboratory research on PE-modified bituminous binder (i.e., instability/inhomogeneity of blend as well as the need for PE grinding). A comparative laboratory study was carried out on dense graded asphalt mixtures containing different amounts of waste plastics (i.e., 0%, 0.25%, and 1.5% by weight of the mixture). The selected asphalt mixes were investigated in terms of workability, linear visco-elastic characteristics, stiffness, strength, resistance to permanent deformation, and moisture sensitivity. Overall, the experimental findings show that the mixes prepared with the dry addition of plastic wastes were able to guarantee almost the same workability and moisture resistance as the reference material while leading to enhanced performance in terms of stiffness and permanent deformation resistance, with better responses for the higher investigated PE dosage.

## 1. Introduction

A circular economy refers to the reuse and recycling of materials aiming for zero waste. To this end, various types of waste and secondary materials can be candidates for use in roads, saving natural resources such as aggregates, fillers, and binders [1]. There is evidence that such waste and secondary materials have been used that have shown a comparable performance to all virgin materials, however, the technology readiness level (TRL) achieved in various parts of the world is varied [2]. The challenge is not the quantities of such materials, but the lack of knowledge, guidelines, and incentives to name some of the barriers for their use.

A considerable amount of plastic waste from construction and demolition waste and municipal solid waste is generated every year. For example, in Europe, each year, 27.1 Mt is produced, 8.4 Mt is recycled, whereas 18.7 is landfilled, incinerated, stored, or exported [2]. This pattern is typical for the fate of plastic waste worldwide.

The use of various types of waste plastics such as polyethylene (PE), polyethylene terephthalate (PET), polyvinylchloride (PVC), and polypropylene (PP) in asphalt mixtures have shown a similar performance as the reference materials, for example, in rutting, moisture resistance, stiffness, and fatigue tests [2,3]. Some waste plastics have reached a technology readiness level of 5–7, indicating that the application is partially or completely industrialized. PE, which is the topic of this study, can be used in asphalt mixtures using the wet process where it is added to the binder before being added to the mixture, or the dry process where it is added directly to the aggregates. Using the dry and the wet processes, better rutting resistance [4,5,6], improved moisture resistance [7,8], stiffness and fatigue have been reported [5,7,9] while some issues regarding low-temperature properties due to increased brittleness have been observed [10].

## 2. Motivation and Research Approach

Given the above background, the RILEM Technical Committee TC-279 WMR investigated the feasibility of recycling waste plastics into asphalt mixtures using interlaboratory tests. To this aim, a specific task group (TG) was first established to assess the chemical and physical properties of asphalt binders modified with waste PE (i.e., wet modification of bitumen). In this framework, more than 10 laboratories worldwide participated in the interlaboratory test program mainly aimed at evaluating the rheological and damage properties of the PE modified binders. The results collected during the above-mentioned comprehensive study [11,12] highlighted encouraging evidence toward the systematic use of waste PE as an effective additive to improve the asphalt binder performance, especially at high service temperatures, provided that waste PE is ground to small particles (around 1 mm maximum size) to enhance the blending with bitumen. However, several issues have been identified in terms of the heterogeneity and variability of the test results. Such low repeatability and reproducibility are likely due to an incomplete melting of the waste plastic into the asphalt binder (a physical rather than a chemical modification was observed), leading to less homogeneous and stable samples with floating plastic particles in the binder matrix, especially at high temperature. Similar issues have also been reported by other researchers testing different waste plastics through different test methods [13,14,15,16,17,18,19,20,21].

To overcome the above-mentioned drawbacks related to the wet modification of the asphalt binder (i.e., the instability/inhomogeneity of the PE-modified binder as well as the need for PE grinding), the dry addition of waste plastics into asphalt mixtures (i.e., the use of waste PE as an aggregate) can be considered as a viable alternative to the wet modification of asphalt binders. In this regard, the present paper illustrates a comprehensive laboratory characterization study of an asphalt mixture prepared with the dry addition of the recycled waste PE tested within the RILEM interlaboratory study cited above. This was to assess the potentialities of the dry modification of asphalt mixtures in overcoming the drawbacks related to the wet process while enhancing the mixture performance. To accomplish this objective, a comparative laboratory study was carried out on dense graded asphalt mixtures containing different amounts of waste plastics (i.e., 0%, 0.25%, and 1.5% by weight of the mixture). A total of 0% plastic waste (i.e., no plastic addition) was selected as a reference mixture for comparison purposes whereas 0.25% plastic corresponded to 5.0% PE by bitumen weight (i.e., the amount used in the previous phase of the research for wet modification of the bitumen [11]). On the other hand, 1.5% PE by weight of mixture was chosen to reproduce a typical amount of “dry additive” into the asphalt mixes. This higher PE content, which was selected by looking at the maximization of waste recycling for environmental reasons, also allowed us to investigate the effect of the plastic content on the mix performance.

The selected asphalt mixes were investigated in terms of workability, linear visco-elastic (LVE) characteristics, stiffness, strength, resistance to permanent deformation, and moisture sensitivity toward a comprehensive picture regarding the contribution of the waste PE. This was to assess whether the addition of such types of waste into the asphalt mixture allows to guarantee the same or even enhanced performance while assuring clear environmental benefits related to its recycling. More details regarding the materials and testing procedures are given in the following.

## 3. Materials and Methods

### 3.1. Materials

Both the reference and the plastic-modified mixtures were prepared using the same raw materials (i.e., aggregates and bitumen) taken from a local asphalt plant. In particular, limestone aggregates typically used in the Italian paving industry were selected. The gradation of available stockpiles (namely 12/20, 8/12, 4/8, 0/4 mm and filler) used to produce the asphalt mixtures are shown in Table 1, whereas the physical properties of such aggregates, complying with common technical specifications, are summarized in Table 2. The physical properties of the plain (non-modified) bitumen with a nominal penetration grade of 70/100 (EN 12591) are listed in Table 3.

Waste polyethylene was supplied by a Swiss recycling center and was the same material used in the RILEM study. This is a secondary waste product obtained during the process of the production of polyethylene pellets that come from waste PE from packaging. This secondary waste is conventionally utilized as fuel in cement plants. Such PE waste comes in the form of a blend of shreds of different shapes and sizes with a density equal to 0.949 Mg/m^3^. Thus, it can be classified as a high-density polyethylene (HDPE), which is typically characterized by a density ranging between 0.941 and 0.965 Mg/m^3^. Its melting temperature was identified in the range between 110 and 124 °C, in accordance with the usual HDPE values (melting point affected by the impurities of the original packaging material). For this reason, it would be able to substantially melt during the typical production processes of asphalt concrete (AC) (in this study, it was carried out at 160 °C) [13]. More details about the PE waste used are illustrated in Figure 1, where a representative picture of the PE shreds is reported along with its granulometric distribution.

### 3.2. Specimen Preparation

Three different kinds of asphalt mixtures were produced to accomplish the objective of the present research. The mix design was carried out considering the same granulometric curve for all mixes, matching the typical Italian technical prescription for an asphalt binder course. To this end, 12/20, 8/12, 4/8, 0/4 mm, and filler stockpiles were combined at 20%, 5%, 30%, 40%, and 5% by weight of mixture, respectively, in order to obtain a granulometric distribution matching the target reported in Table 4. As mentioned earlier, two different PE contents were assessed by adding the plastic shreds using the dry process, in percentages of 0.25% and 1.5% by mix weight, hereafter labelled PEs02 and PEs15, respectively. An unmodified mixture not containing PE shreds, hereafter coded REF, was manufactured for comparison purposes. The different densities and dosages of all constituents were considered to adjust the maximum theoretical density and calculate the bulk density of each mixture with a target void content equal to 4.5%. A bitumen dosage of 5% by aggregate weight was obtained based on a preliminary Marshall mix design of the reference mixture (i.e., the mixture without plastic waste). The aggregate gradation and bitumen content were not adjusted with the addition of the waste PE in order to limit the variables to be considered. Details about the mix design are given in Table 5.

Asphalt mixtures were prepared by pre-heating the aggregates at 160° for 3 h, then adding (when included) the selected quantities of PE shreds, which were stored at room temperature, to the hot aggregates. The PE and hot aggregates were mixed for 1 min; hot bitumen (160 °C) was then added and blended mechanically by stirring the mixture for 2 min to obtain an adequate mix of homogeneity as assessed visually. Finally, the filler at ambient temperature was added to the hot batch for a 1 min long final mixing.

The 80 mm thick cylindrical specimens with a diameter of 150 mm were produced using the Superpave gyratory compactor (SGC) at a fixed height by applying a rotation speed of 30 rpm and a pressure of 600 kPa with an inclination angle of the mold equal to 1.25° (EN 12697-31). Then, the gyratory samples were sawed to produce the testing samples with a height of 40 mm. It is worth highlighting that the compaction of specimens using the SGC allowed for the construction of the compaction curves (i.e., voids vs. number of gyrations), thus providing fundamental information regarding the workability of mixtures containing waste plastics. Asphalt concrete slabs (400 mm long, 300 mm wide, and 50 mm thick) were also prepared at the selected target voids through a roller compactor (EN 12697-33) to obtain, by sawing, 50 mm wide prismatic specimens for the flexural tests.

### 3.3. Testing Methods

#### 3.3.1. Flexural Cyclic Tests

Prismatic samples were subjected to cyclic frequency sweep tests in the 4-point bending (4PB) configuration (EN 12697-26/Annex B). These non-destructive tests were replicated at different temperatures *T* (10, 20, 30, 40 °C) to achieve a comprehensive picture of the LVE characteristics of the investigated materials (four replicates for each AC at each temperature). A pre-conditioning time of at least 4 h was guaranteed at the selected temperature. Tests were performed by applying strain-controlled loads in a sinusoidal configuration (50 μstrain amplitude) at frequencies *f* of 0.1, 0.2, 0.5, 1, 2, 5, 10 20 Hz (100 load repetitions for each *f*). A total of 100 load cycles at 0.1 Hz were applied again at the end of each test and compared to the corresponding findings measured at the beginning of the tests to verify that the specimens had not experienced any damage during the frequency sweeps. The complex modulus *|E*|* (stiffness) and phase angle *Φ* (stress–strain lag due to viscoelasticity) were calculated through the force measured with a load cell and the corresponding displacement using a transducer mounted at the middle section of the beam specimen. Stiffness master curves were constructed thanks to the time–temperature superposition principle (TTSP), shifting the data series through specific shift factors at the reference temperature of 20 °C through the Williams–Landel–Ferry (WLF) equation [22]. In the case of the PE-modified mixes, the applicability of such a principle was preliminarily evaluated by observing the experimental data plotted in the Black diagram (|*E**| vs. *Φ*), considering that the validity of TTSP, which allows us to consider the material as thermo-rheologically simple, normally results in a uniquely aligned trend in such Black space [23]). The well-known NCHRP rheological universal model [24] was used to produce the master curves of |*E**|, modeling the curves through the following equation:*E** = *E_e_** + {(*E_g_* − E_e_**)/[(1 + *f_c_*/*f’*)^k^]*^me^*^/k^}(1)
where

*E_e_** is the equilibrium complex modulus, at *f* → 0;*E_g_** is the glass complex modulus, at *f* → ∞;*f_c_* is the location parameter, with dimensions of frequency (frequency at which *Φ* = 45°);*f’* is the reduced frequency, function of both temperature and strain;*k*, *m_e_* are the shape dimensionless parameters.

#### 3.3.2. Indirect Tensile Stiffness Modulus Tests

The stiffness characteristics of the mixtures were also investigated in indirect tensile configuration on cylindrical samples (IT-CY) according to EN 12697-26/Annex C. After a pre-conditioning time of at least 4 h at the test temperature of 25 °C, IT-CY tests were carried out in strain-controlled mode, applying five haversine pulses (along the vertical diameter). The corresponding horizontal deformation was monitored by two linear variable displacement transducers mounted opposite one another in a rigid frame clamped to the samples. Loads were characterized by a rise time equal to 124 ms (time to apply the load from zero to peak) and a target peak horizontal strain of 7 μm (10 pre-conditioning pulses were given before the test to set the loading cell). Tests were performed on twelve replicates for each mixture, applying the pulses on two perpendicular diameters (i.e., rotating the specimen through 90° about its horizontal axis between the two measurements). Stiffness values *E* were calculated according to the referenced standard from the force peak values applied (as the average of the two diameters tested for each sample).

#### 3.3.3. Repeated Load Axial Tests

Uniaxial cyclic compression tests (CCT) with confinement (EN 12697-25/Method A1) were executed on cylindrical specimens to determine the resistance of bituminous mixtures to permanent deformation. Such destructive tests allowed us to assess the rutting potential of asphalt concrete (two replicates for each mixture were tested). Cyclic axial load pulses were applied in stress-controlled mode at the selected test temperature (after a pre-conditioning time of at least 4 h). Two stress levels *σ* were investigated (i.e., 100 kPa and 300 kPa) to assess the applied stress sensitivity whereas test temperature *T* was fixed at 60 °C. Load was applied with a frequency of 0.5 Hz (1 s loading time at the stress level *σ*, followed by 1 s rest period) up to 3600 loading cycles. The 150 mm diameter specimens were centrally loaded in the axial direction with a circular plate having a diameter of 100 mm to reproduce the confinement of field conditions (the ring of material non-directly loaded simulates the real-scale confining action due to surrounding pavement). Rutting behavior was evaluated by constructing the relationship between the cumulative axial strain *ε_ca_* and the number of applied cycles *n* (average of the two replicates); then, the creep rate parameter (slope of the quasi-linear part of *ε_ca_* vs. *n* curve) was estimated as a key parameter to assess the resistance to permanent deformation of the tested mix (Figure 2). The final creep modulus *E_f_* (i.e., the ratio between the stress level and the corresponding cumulative strain at the end of the test) was also calculated for a better understanding of the stress sensitivity of the studied mixes.

#### 3.3.4. Indirect Tensile Strength and Fracture Tests

Indirect tensile configuration was also used to investigate the fracture resistance of the mixtures. To this end, static indirect tensile strength (ITS) tests (EN 12697-23) were carried out at 25 °C on three cylindrical sample replicates (at least 4 h of pre-conditioning time was also guaranteed in this case). According to standard test conditions, a vertical strain rate equal to 51 mm/min was applied while recording the load evolution over time. The ITS parameter was calculated from the peak load measured at the sample’s failure. The dissipated energy to fracture (*E_ITS_*) (i.e., the area under the force-displacement (F–s) curve up to the peak failure) was also calculated (Figure 3).

In the second case, samples were subjected to pulse load repetitions to assess the fracture potential due to dynamic phenomena. Cylindrical samples were subjected to indirect tensile fatigue (ITF) tests performed according to the British standard BS DD ABF. A cyclic pulse load, with a repetition period of 1.5 s and a rise time of 124 ms, was applied in stress-controlled mode in the vertical direction (100 kPa and 200 kPa were selected as the input stress levels *σ*). The temperature was set again to 25 °C (with pre-conditioning of at least 4 h). One specimen for each test configuration (mixture and stress level) was tested. The output was expressed in terms of the resilient strain amplitude (*ε_res_*) over cycles *N*. The energy ratio (*E_ratio_*), expressed as *N × σ/**ε_res_*, was then plotted to calculate the number of cycles to failure *N_f_* (peak of the *E_ratio_* vs. *N* curve), as described in Figure 4. This allowed us to simultaneously consider the effect on both the tensile strength and the strain tolerance.

#### 3.3.5. Moisture Resistance Assessment

The ITS tests were also carried out on cylindrical specimens preliminarily subjected to wet conditioning in order to investigate the AC susceptibility to moisture (three wet and three dry replicates for each mixture) by comparing the results with those in dry conditions (i.e., the same samples described in section above). Before the ITS tests, samples were placed in a temperature-controlled bath at 60 ± 1 °C for 24 ± 1 h according to the AASHTO 283 standard. Then, they were air conditioned at 25 °C for 4 h before testing. The moisture resistance was considered by calculating the indirect tensile strength ratio (ITSR) as the percentage ratio of the ITS values of the wet conditioned samples and the non-conditioned ones. It is worth noting that due to the limited amount of material available, the specimens used for the assessment of ITSR were not compacted at a higher void content as required by the standard procedure.

## 4. Results and Analysis

### 4.1. Compactability

The use of SGC allowed for the assessment of the workability of the investigated mixtures by analyzing the experimental findings recorded during compaction (i.e., the increase in volumetric bulk density was calculated thanks to the continuous reading of the height). According to EN 12697-10, compactability curves were then obtained by linear regression of the variation of the air voids *v_%_* (calculated as 100%-degree of compaction, which is the percentage ratio between the measured bulk density and the theoretical maximum density) against the natural logarithm of the number of gyrations *ln(n_g_)*, which represents the compaction energy. The obtained trends for each of the six specimens compacted using SGC for each mix are shown in Figure 5a–c and were used to calculate the *compactability K*, that is, the slope of the *v_%_* vs. *ln(n_g_)* regression lines.

As shown in Figure 5d, the three investigated materials were characterized by the same average compactability with limited variability, suggesting no significant effects of waste PE addition on the asphalt mixture workability at the investigated production conditions, even when dosed at a high rate (1.5% by mix weight). Such an experimental finding, also considering that the mix design was not adjusted to account for the presence of the PE, is rather promising, since it is possible to assume no detrimental impact on the material preparation and laying during field construction (i.e., with no need of specific changes/expedients such as the adoption of higher construction temperatures or the use of additives to achieve the desired workability. This fundamental outcome is in accordance with recent research that has reported similar compactability for the AC14 asphalt mixtures prepared by replacing part of the coarse or the fine aggregate with recycled plastic, provided that the waste plastic is dosed at maximum 2% by mix weight [2].

### 4.2. LVE Properties

The LVE properties of the studied mixtures were investigated through frequency sweeps tests in the 4PB configuration. The experimental findings are reported in Figure 6 and Figure 7 and depict the measured data in the Black space (norm of the complex modulus |*E*|* vs. the corresponding phase angle *Φ*) and Cole–Cole plot (the viscous component *E*_2_ vs. the elastic component *E*_1_ of the complex modulus), respectively.

Such results clearly show the contribution of the addition of waste PE on the LVE properties of the selected asphalt mixtures. In particular, it is worth noting that the lowest dosage of PE led to a slight change in the LVE characteristics, limited to a small stiffening at the high-temperature/low-frequency domain, whereas a higher amount of waste plastic significantly affected the rheological properties across the whole investigated time–temperature domain. Hence, according to recent literature [9], the PEs15 mix is characterized by a clear shift toward a more elastic response in the whole domain along with a considerable stiffening effect at the high-temperature/low-frequency domain. This promising effect discloses a distinctly enhanced performance against permanent deformation and also suggests unvaried (or even higher) properties at lower temperatures since a more elastic response (i.e., lower phase angle) is associated with almost the same stiffness (i.e., norm of the complex modulus). Such modification of the rheological response of the asphalt mixture also indicates that PE does not act as a simple solid addition to the mix, but it is likely to hypothesize a significant degree of melting of the waste plastic, which could promote a certain chemical interaction with the asphalt binder, especially at the highest recycling rate. Indeed, it must be advised that such modification did not alter the overall thermos-rheological behavior of the mixtures (all mixes remained thermos-rheologically simple in light of the continuous-aligned trends found in Black spaces [25]).

In order to evaluate the significance of the measured LVE characteristics, Table 6 reports some of the statistics about the test repeatability found in the laboratory and evaluated according to the ASTM C670 standard. The difference between the highest and lowest experimental data for each frequency Δ_max−min_ was checked with respect to the single-operator precision limits with a confidence level of 95%. In this regard, Table 6 reports, for each test frequency, the maximum difference recorded among all of the tested temperatures. The acceptance limits indicated in Table 6 were calculated by multiplying the standard deviation *1s* of the four replicate values by the related statistical multiplier given by ASTM C670 for a group of four samples.

It is worth noting that all values fell within the acceptance limits, therefore, the tests can be considered as reliable and data can be consistently adopted. Moreover, similar variabilities were found for the REF and PE-modified mixes, suggesting that PE particles within mixtures did not cause great inhomogeneity, in contrast with the heterogenous results observed at the binder scale [11,12].

For the sake of completeness, Figure 8, Figure 9 and Figure 10 report the frequency sweep test results in terms of the master curves of the complex shear modulus *|E*|* constructed according to the models already described in Section 3.3.1, whereas Figure 11 summarizes all of the results in a single graph to allow for a prompt data comparison. In this regard, the key parameters of the master curves are reported in Table 7 whereas Figure 12 shows the comparison between the models. Results confirmed the improvement in the high temperature properties for the PE modified mixtures, especially in the case of PEs15. Master curves also highlighted the lower stiffness at high reduced frequencies (i.e., low temperatures) for the mix with 1.5% plastic shreds, while almost unaltered behavior could be observed for the lowest PE dosage. This is reflected in the flattening of the curve and a lower time–temperature dependence for PEs15.

### 4.3. Stiffness

The stiffness moduli *E* determined through the IT-CY non-destructive tests are presented in Figure 13. The graph reports all 36 measured values (12 specimens tested for each selected material) and highlights the corresponding average values for each mixture. Again, according to previous studies [15,16,26], the presence of the waste PE led to a clear increase in the stiffness moduli at intermediate service temperatures; in particular, it was possible to observe that the higher the PE content, the higher the stiffness modulus, with a 114% increase in the case of the PEs15 mix containing 1.5% waste plastic. Given the fact that all three mixtures were compacted at fixed void content, the observed stiffening effect can likely be attributed to the presence of PE and its interaction with the aggregate solid skeleton as well as with the bituminous binder, confirming the analyses carried out based on the LVE characteristics measured in the 4PB configuration.

To evaluate the significance of the measured differences in the collected data for the three mixtures, a statistical evaluation through a one-way Analysis of Variance (ANOVA) at a 95% confidence level was carried out. The results of such statistical study are summarized in Table 8, showing that the increase in stiffness due to the dry addition of waste PE is statistically significant; moreover, a different dosage of PE in the investigated range (i.e., from 0.25% to 1.5% by the weight of the mix) significantly affect the stiffness properties of asphalt mixtures.

### 4.4. Permanent Deformation Resistance

The CCT results at the selected stresses are presented in Figure 14 in terms of the cumulative axial strain (to obtain a suitable comparison, data were expressed in dimensionless terms, dividing the punctual axial strain by the initial one *ε_ca_*/*ε_ca,in_* on the y-axis). To quantify the accumulated deformations in absolute terms, final cumulative strains (*ε_ca,f_*) are given in Table 9, together with the calculated creep rate parameters (*f_c_*) and creep modulus (*E_f_*).

Overall, the experimental findings show that a clear increase in permanent deformation resistance at high-service temperature was achieved by adding the selected waste PE to the reference asphalt mixture, according to the existing literature [5,14,15,16,27,28]. In particular, about a 30% reduction in the final cumulative axial strain was measured at a 100 kPa stress level, where the plastic mixtures demonstrated a similar performance, even if the higher dosage of plastic shreds led to a significantly lower creep rate.

Obviously, the specimens subjected to a higher stress level (i.e., 300 kPa) suffered higher permanent deformation levels regardless of the material; however, this more detrimental test condition differentiated even more the behavior of the tested mixtures demonstrating a lower stress sensitivity for the PE mixes. Indeed, the reference material tested at 300 kPa showed the tertiary phase of the evolution of the permanent strain (i.e., an increasing rate of deformation with loading cycles) toward the physical failure, whereas the mixtures containing plastics were still in the quasi-linear phase of their *ε_ca_* vs. *n* curves. Analogous results have been reported by other researchers who performed similar tests at both 40 °C and 50 °C in dry and wet conditions for mixtures containing 5% plastic waste by weight of bitumen [5].

Moreover, the effect of PE dosage was even more evident since PEs15 was able to guarantee about 60% and 90% lower creep rate with respect to the PEs02 and REF materials, respectively; a significantly lower final cumulative axial strain was also observed for PEs15 (21% and 56% reduction with respect to the PEs02 and REF mixes, respectively), leading to an even greater increase in creep stiffness modulus.

### 4.5. Fracture Resistance

The fracture resistance of the studied mixtures was assessed through indirect tensile tests in both the static and dynamic loading modes. In this regard, Table 10 reports the measured indirect tensile strength (ITS) of all of the tested specimens along with the corresponding average values for each material, whereas Figure 15 shows the registered force–displacement trend (*F* vs. *s*) and the related dissipated energy *E_ITS_* up to the peak failure as described in the section above. Figure 16 reports the same experimental data in a single graph for the sake of an effective material comparison.

The experimental findings demonstrated a slight increase in strength with the addition of 5% waste plastic by bitumen weight (i.e., PEs02 mix), whereas the higher PE amount led to a significantly higher tensile strength of the asphalt concrete, confirming the hypothesized strengthening of the asphalt mastic due to the chemo-physical interaction between the asphalt binder and the plastic particles [10]. However, notwithstanding the higher peak forces observed for the PE mixtures, very similar energy-to-failure values were found for the tested materials, suggesting a possible decrease in ductility for higher amounts of plastics, as already found by other researchers [2,14,28]. This fact could reflect in possible issues against the low temperature cracking resistance of such a material, therefore, a specific study is suggested for further analysis.

The dynamic ITF tests carried out by applying a cyclic pulse loading in stress-controlled mode can provide further indications. In this sense, Figure 17 presents the relationships between the resilient strain amplitude and the number of loading cycles at 25 °C at the selected stress levels (i.e., 100 and 200 kPa) whereas Figure 18 plots the energy ratio (*N × σ/**ε_res_*) against the number of cycles, also indicating the number of cycles to failure *N_f_* for all of the mixtures represented by the vertical dotted lines.

As can be seen, the tests conducted in dynamic mode confirmed the enhanced tensile resistance that was achievable thanks to the dry inclusion of the selected waste plastics into the asphalt mixtures without negative aspects related to a possible reduction in ductility. Indeed, PEs02 was able to withstand five times higher loading cycles whereas the addition of 1.5% plastic by mix weight led to about a 20 times higher fracture resistance; in addition, both plastic mixtures were characterized by a rather comparable strain at failure with respect to the control mixture, even if a slight reduction could be detected in the case of a high plastic content. In this sense, Giustozzi et al. [2] also highlighted the capability of dissipating energy via deformation of the mixtures prepared by partially replacing natural aggregates with recycled plastic. Moreover, in the investigated range of stresses, it is worth noting that there was a very similar stress sensitivity for all of the materials investigated in the present study.

### 4.6. Moisture Resistance

Figure 19 shows the ITS of the wet-conditioned samples and the corresponding ITSR values to evaluate the effect of the waste PE shreds on the moisture resistance of the investigated asphalt mixtures.

First, it is interesting to note that the ITS of the wet conditioned samples depict the same behavior observed in the dry condition, with a slightly higher strength for PEs02 and a more significant increase in strength for the higher dosage of plastic. This fact reflects in ITSR indices very close to 100%, indicating practically no detrimental effect of water conditioning for both the reference mixture and the materials containing plastic. Thus, it can be stated that, similarly to the outcomes of recent studies [15,16,27], a plastic addition up to 1.5% by the mix weight did not negatively affect the moisture resistance under the investigated standard test conditions. Further research subjecting the investigated materials to more severe water conditioning can be suggested to confirm this outcome.

## 5. Conclusions and Prospects for Further Studies

The present research study evaluated the feasibility of producing asphalt mixtures with the dry addition of waste polyethylene (PE). To this aim, a comprehensive experimental program was carried out in the laboratory to analyze the effect of different dosages of PE (i.e., 0.25% and 1.50% by the weight of the mixture). A reference material without the addition of the plastic waste was also investigated for comparison purposes.

Based on the experimental findings, the main conclusions can be summarized as follows:The addition of waste PE up to 1.5% by mix weight did not affect the asphalt mixture workability at the investigated production conditions.PE waste at the highest recycling rate used in this investigation significantly influenced the rheological properties of the asphalt mixture with a clear shift toward a more elastic response in the whole investigated time–temperature domain along with a considerable stiffening effect at the high-temperature/low-frequency domain.Significantly higher stiffness and permanent deformation resistance were obtained in the case of the mixtures containing recycled waste plastic; again, the higher the PE dosage, the better the material response.A proportional increase in the strength and fracture resistance with the corresponding addition of plastic waste was also observed in indirect tensile tests, suggesting the strengthening of the asphalt mastic due to the chemo-physical interaction between the asphalt binder and the plastic particles.Moisture resistance of the investigated asphalt mixture was not negatively affected by the presence of waste PE.

Thus, the experimental results achieved during this first phase of the research provide promising evidence toward the successful use of such plastic waste into asphalt mixtures by using the dry method. Further research is needed to better understand the chemo-physical interaction between the plastic particles and asphalt binder at the microstructure level in order to generalize the particular indications coming from the present study and provide precise guidelines for mixture preparation and control. Moreover, a specific assessment of the low-temperature properties of PE-modified asphalt mixtures is recommended in order to exclude and/or limit the possible drawbacks related to a prospective reduction in ductility. Round robin inter-laboratory investigations within the framework of the RILEM TC strengthen these findings and are underway. Field trial sections should be constructed and monitored to validate the laboratory findings.

## Figures and Tables

**Figure 1 materials-15-04739-f001:**
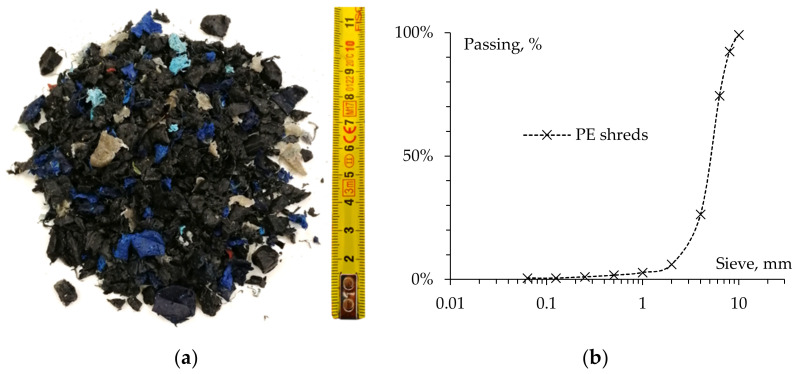
The polyethylene waste used in this study: (**a**) photograph; (**b**) gradation of sample.

**Figure 2 materials-15-04739-f002:**
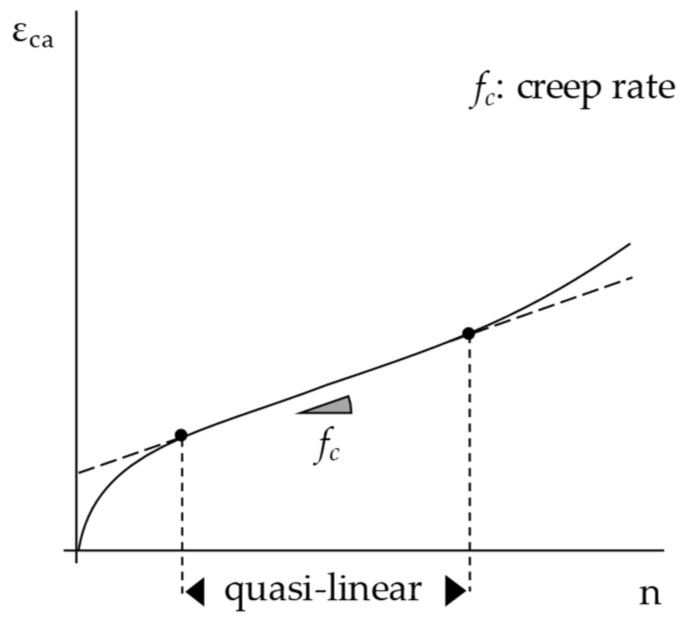
The creep rate parameter determined through the CCT test.

**Figure 3 materials-15-04739-f003:**
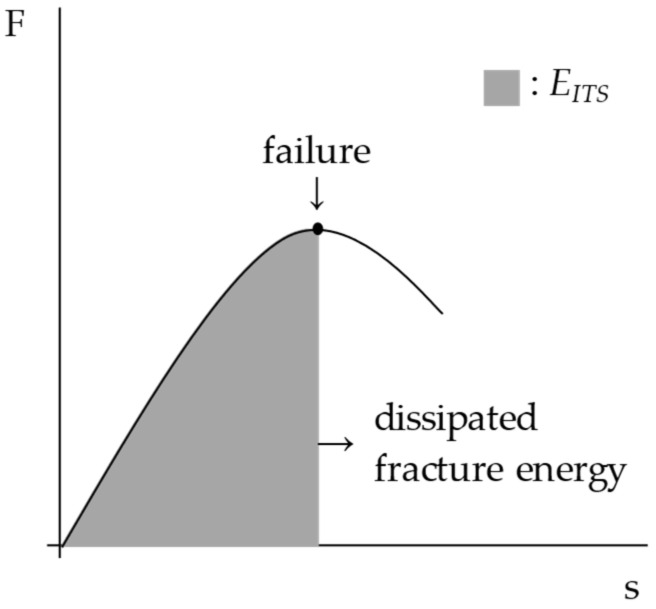
Calculation of the dissipated energy to fracture through the ITS test.

**Figure 4 materials-15-04739-f004:**
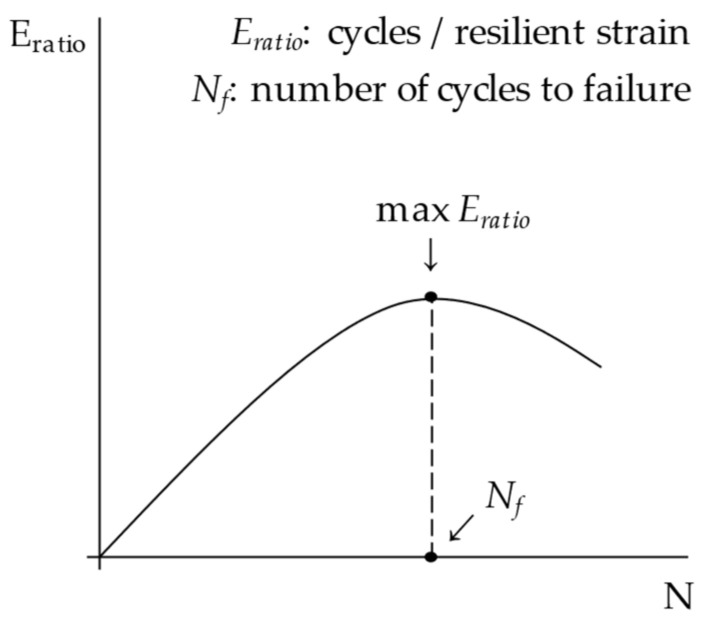
The calculation of the number of cycles to failure through the ITF energy ratio.

**Figure 5 materials-15-04739-f005:**
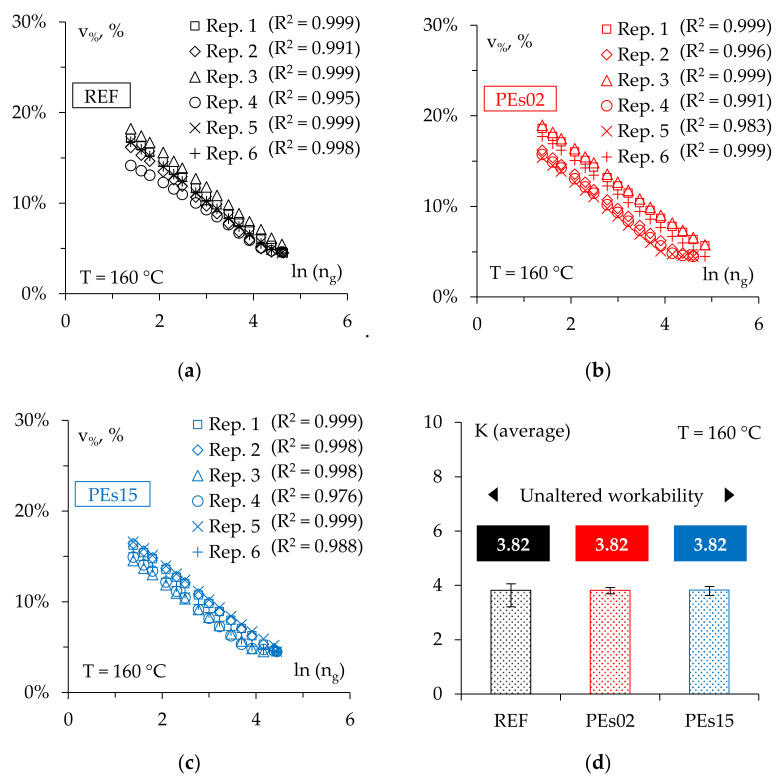
The compaction curves ((**a**) REF; (**b**) PEs02; (**c**) PEs15) and the workability (**d**) of the mixes.

**Figure 6 materials-15-04739-f006:**
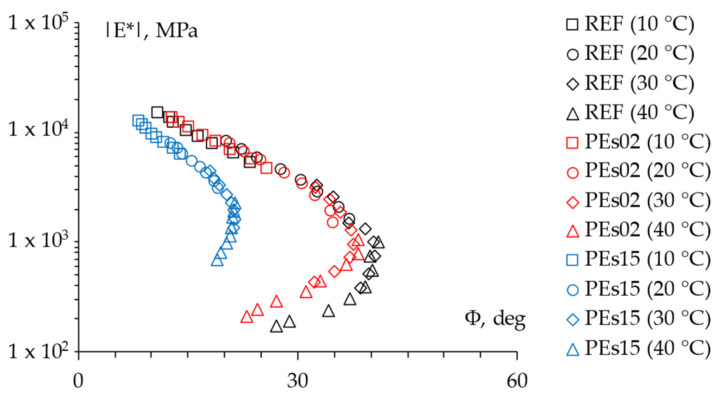
The 4PB experimental findings: Black diagram.

**Figure 7 materials-15-04739-f007:**
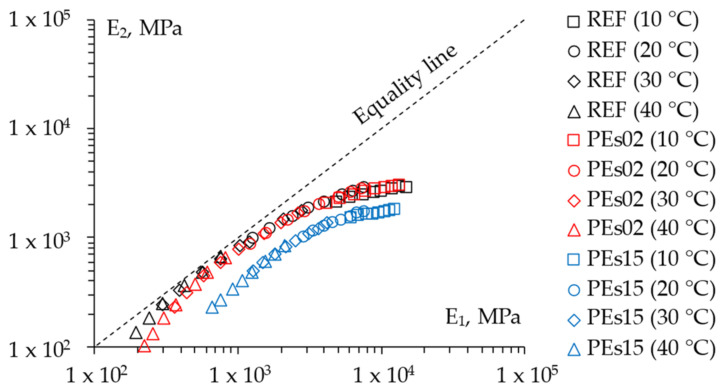
The 4PB experimental findings: Cole–Cole plot.

**Figure 8 materials-15-04739-f008:**
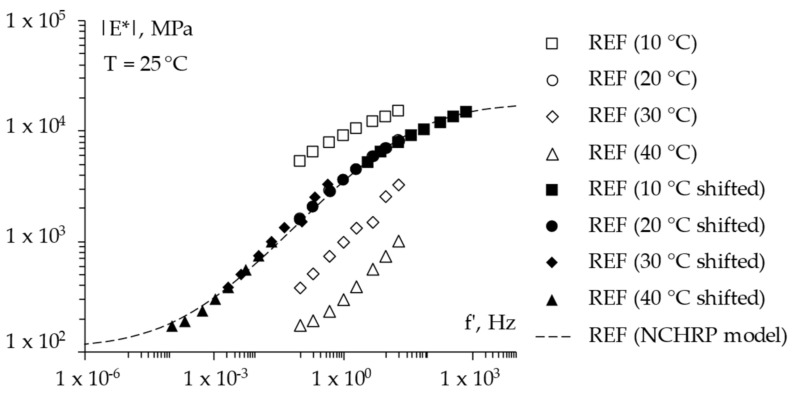
The frequency sweep results (master curve) of the *REF* mixture.

**Figure 9 materials-15-04739-f009:**
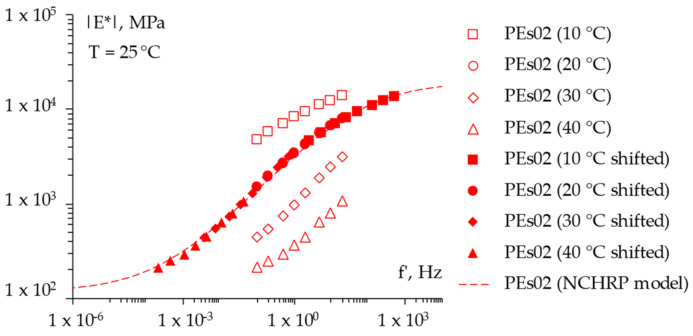
The frequency sweep results (master curve) of the *PEs02* mixture.

**Figure 10 materials-15-04739-f010:**
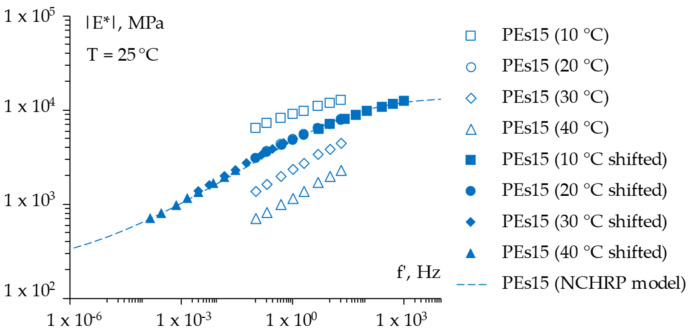
The frequency sweep results (master curve) of the *PEs15* mixture.

**Figure 11 materials-15-04739-f011:**
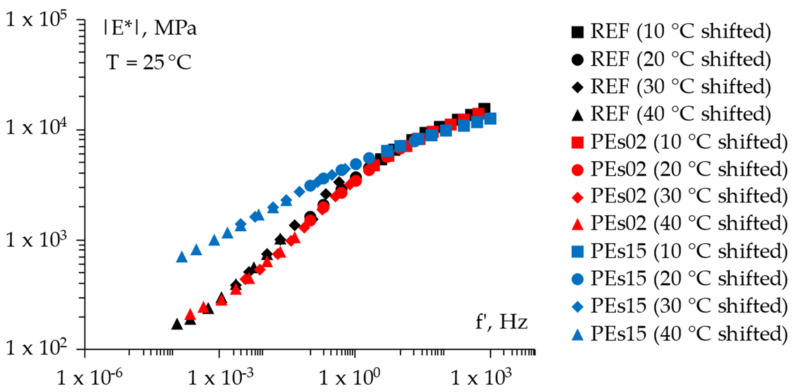
A comparison of the *E** master curves (all mixtures).

**Figure 12 materials-15-04739-f012:**
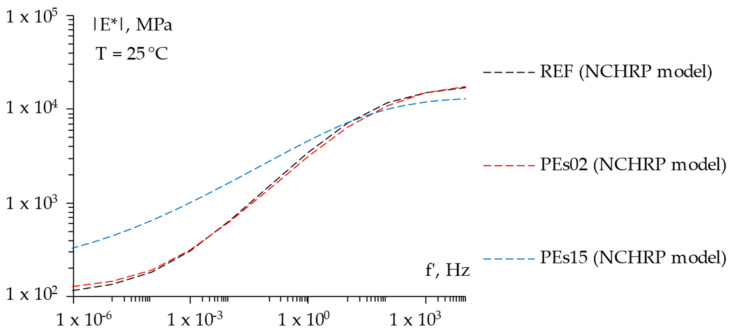
The NCHRP models for the studied mixtures.

**Figure 13 materials-15-04739-f013:**
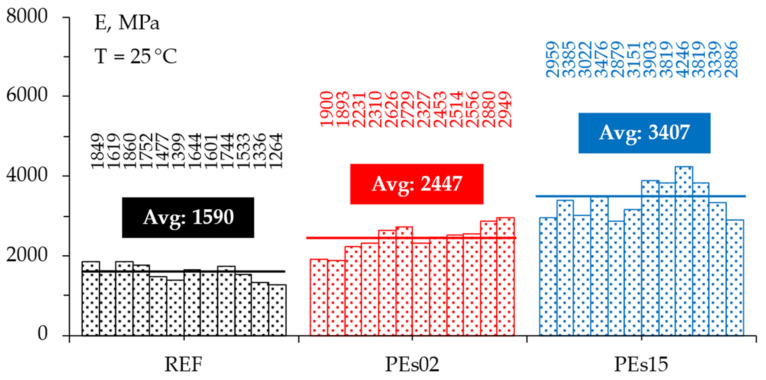
The stiffness modulus determined through the IT-CY tests.

**Figure 14 materials-15-04739-f014:**
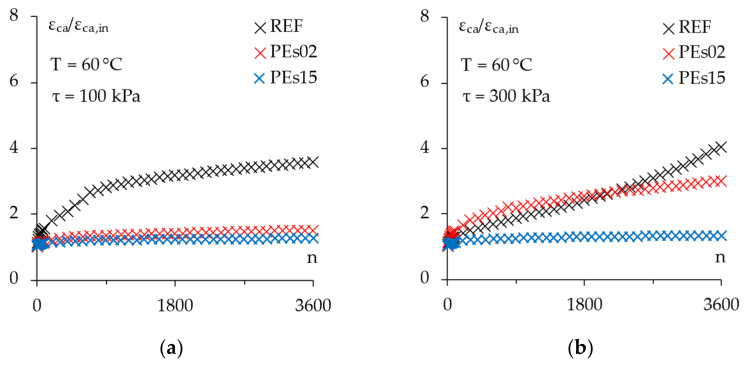
The CCT results. Evolution of the cumulative axial strain at: (**a**) 100 kPa; (**b**) 300 kPa.

**Figure 15 materials-15-04739-f015:**
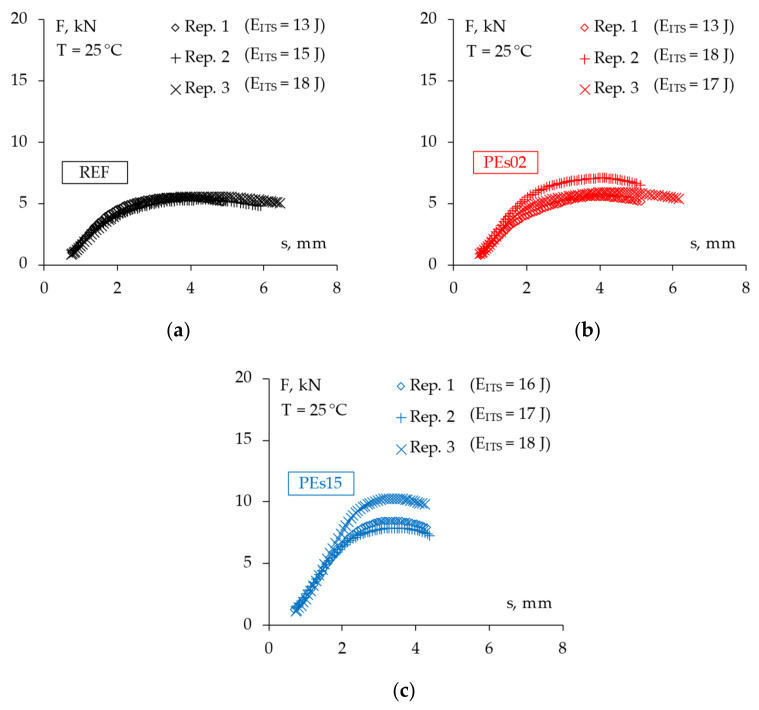
The force–displacement trend from the ITS tests for all mixtures: (**a**) REF; (**b**) PEs02; (**c**) PEs15.

**Figure 16 materials-15-04739-f016:**
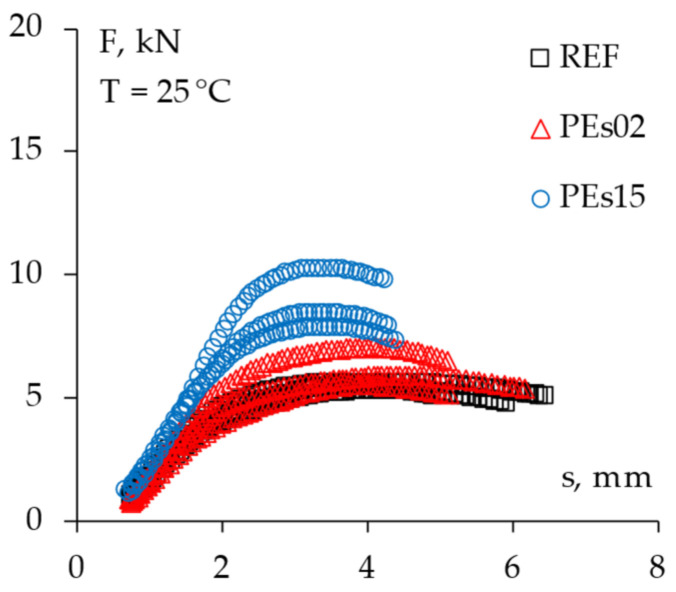
The comparison of the force–displacement trends from the ITS tests (all mixtures).

**Figure 17 materials-15-04739-f017:**
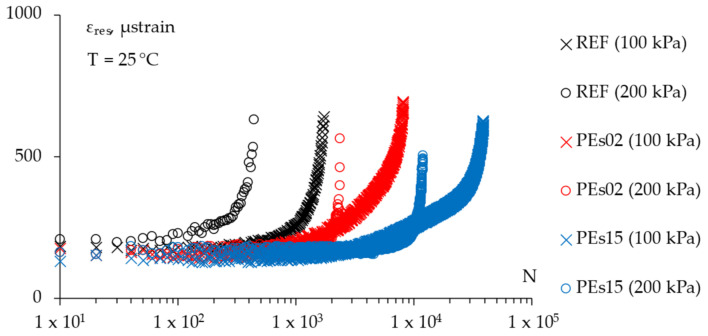
The ITF test results: evolution of resilient strain amplitude on the test cycles.

**Figure 18 materials-15-04739-f018:**
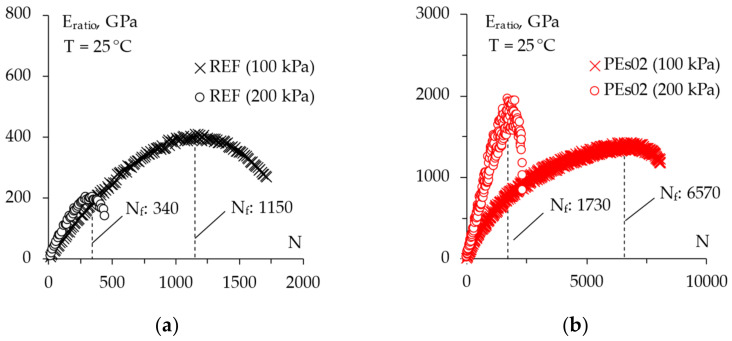
The ITF test energy ratio for the tested mixtures: (**a**) REF; (**b**) PEs02; (**c**) PEs15.

**Figure 19 materials-15-04739-f019:**
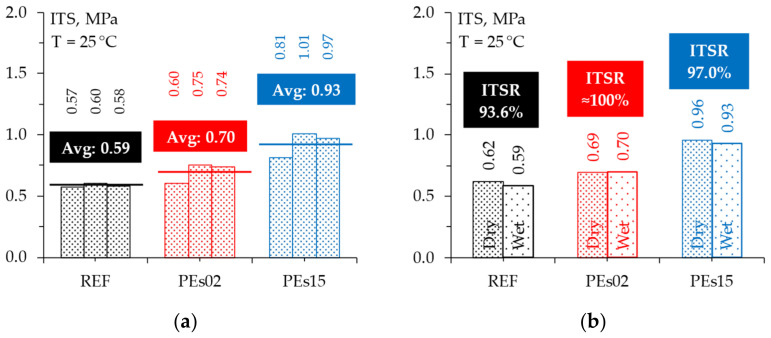
The moisture resistance: (**a**) ITS of wet-conditioned samples; (**b**) ITSR values.

**Table 1 materials-15-04739-t001:** The grade compositions of the available aggregates.

Sieve Size, mm	Passing, %				
	12/20	8/12	4/8	0/4	Filler
25	100	100	100	100	100
20	97.9	100	100	100	100
16	82.2	100	100	100	100
12.5	52.4	97.9	100	100	100
10	24.6	70.7	100	100	100
6.3	0.6	4.8	63.8	100	100
4	0.6	0.7	6.8	95.8	100
2	0.6	0.7	0.1	72.2	100
0.5	0.6	0.7	0.1	33.3	100
0.25	0.6	0.7	0.1	23.2	97.3
0.063	0.6	0.7	0.1	9.3	57.7

**Table 2 materials-15-04739-t002:** The physical properties of the selected limestone aggregates.

Property	Standard	Unit	12/20	8/12	4/8	0/4	Filler
Flakiness index	EN 933-3	%	9	7	10	-	-
Shape index	EN 933-4	%	-	16	11	-	-
Sand equivalent	EN 933-8	%	-	-	-	82	-
LA coefficient	EN 1097-2	%	21	-	-	-	-
Particle density	EN 1097-6	Mg/m^3^	2.71	2.72	2.69	2.66	-
Rigden voids	EN 1097-4	%	-	-	-	-	32.7
Particle density	EN 1097-7	Mg/m^3^	-	-	-	-	2.74

**Table 3 materials-15-04739-t003:** The physical properties of plain bitumen.

Property	Standard	Unit	Value
Penetration at 25 °C	EN 1426	0.1 mm	78
Softening point	EN 1427	°C	42
Ductility	ASTM D113	cm	>100

**Table 4 materials-15-04739-t004:** The mix gradation and Italian specification limits for an AC binder course.

Sieve Size, mm	32	16	10	4	2	0.5	0.25	0.063
	Passing, %							
Min. envelop limit	100	90	73	45	28	16	11	4
Max. envelop limit	100	100	85	56	38	24	18	8
Mix gradation	100	96.4	83.5	45.5	34.1	18.5	14.3	6.8

**Table 5 materials-15-04739-t005:** The mix design: the proportions and characteristics of the produced mixtures.

Mix Designation	Aggregate	Bitumen	PE
	REF		
Content by agg. weight, %	100.00	5.00	-
Content by mix weight, %	95.24	4.76	-
Content by mix volume, %	88.18	11.82	-
Max. density, Mg/m^3^	2.506		
Target voids, %	4.5		
	PEs02		
Content by agg. weight, %	100.00	5.00	0.26
Content by mix weight, %	95.00	4.75	0.25
Content by mix volume, %	87.61	11.74	0.65
Max. density, Mg/m^3^	2.496		
Target voids, %		4.5	
	PEs15		
Content by agg. weight, %	100.00	5.00	1.60
Content by mix weight, %	93.81	4.69	1.50
Content by mix volume, %	84.77	11.36	3.87
Max. density, Mg/m^3^	2.446		
Target voids, %	4.5		

**Table 6 materials-15-04739-t006:** The 4PB test repeatability: single-operator precision for the stiffness modulus and phase angle.

Quantity	Frequency, Hz							
	0.1	0.2	0.5	1	2	5	10	20
	**REF**							
|*E**| Δ_max − min_	914	1006	1261	1496	1666	1979	2204	2436
Limits	1328	1386	1774	2060	2283	2730	3180	3661
*Φ* Δ_max − min_	2.90°	2.70°	2.30°	2.71°	3.66°	2.81°	2.06°	2.90°
Limits	4.21°	3.82°	3.12°	3.56°	4.85°	3.97°	2.71°	3.71°
	**PEs02**							
|*E**| Δ_max − min_	903	1094	1042	1532	1766	2077	2210	2936
Limits	1226	1540	1571	2222	2510	2924	3172	3967
*Φ* Δ_max − min_	5.66°	2.07°	2.25°	9.34°	2.25°	3.61°	1.97°	2.49°
Limits	7.24°	2.74°	3.07°	13.18°	3.11°	4.69°	2.85°	4.26°
	**PEs15**							
|*E**| Δ_max − min_	786	573	655	737	831	931	1053	1120
Limits	1044	835	965	1060	1155	1272	1429	1468
*Φ* Δ_max − min_	1.87°	2.69°	2.92°	2.36°	2.22°	2.42°	3.53°	3.49°
Limits	2.51°	3.76°	3.94°	3.30°	3.16°	3.45°	4.57°	4.77°

**Table 7 materials-15-04739-t007:** The parameters of the NCHRP fitting model for the complex modulus master curves.

Mixture	Parameter					Reliability
	*f_c_*, Hz	k	*m_e_*	*E_e_*, MPa	*E_g_*, MPa	*R^2^*
REF	30.17	0.446	0.434	107	18,120	0.9963
PEs02	54.17	0.411	0.420	118	19,546	0.9975
PEs15	56.66	0.459	0.254	189	13,645	0.9959

**Table 8 materials-15-04739-t008:** The results of the ANOVA performed on the measured IT-CY stiffness moduli.

Mixes Comparison	Significant?	F	F-Crit.	*p*-Value
REF vs. PEs02	Yes	58.116	4.300	1.30 × 10^−7^
REF vs. PEs15	Yes	162.558	4.300	1.23 × 10^−11^
PEs02 vs. PEs15	Yes	34.547	4.300	6.49 × 10^−6^

**Table 9 materials-15-04739-t009:** The CCT results: initial and final cumulative axial strain and creep rate of the mixtures.

Mixture	100 kPa			300 kPa		
	*ε_ca,f_* [µstrain]	*f_c_* [µstrain/cycle]	*E_f_* [MPa]	*ε_ca,f_* [µstrain]	*f_c_* [µstrain/cycle]	*E_f_* [MPa]
REF	17068	1.483	5.86	54432	8.832	5.51
PEs02	11999	0.905	8.33	30213	2.801	9.93
PEs15	11246	0.488	8.89	23787	1.135	12.61

**Table 10 materials-15-04739-t010:** The indirect tensile strength (ITS) values for each mixture.

Mixture	ITS, MPa			
	Rep. 1	Rep. 2	Rep. 3	Average
REF	0.67	0.58	0.62	0.62
PEs02	0.64	0.76	0.68	0.69
PEs15	0.95	0.81	1.11	0.96

## Data Availability

The data presented in this study are available on request from the corresponding author. The data are not publicly available because they are a part of on-going research.

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
