# Peer review of "Dry Addition of Recycled Waste Polyethylene in Asphalt Mixtures: A Laboratory Study"

_materials, 2022, doi:10.3390/ma15144739_

Round 1

Reviewer 1 Report

The authors presented interesting studies on use of plastic waste such as polyethylene (PE) in asphalt mixtures. In showed investigation PE waste in amount of 0.25% and 1.5% by weight of the mixture have been applied. Essentially, the experiments have been planned and developed in accordance with the rules, and the results are well, explicitly and correctly presented in the paper. Every observed phenomenon is very well described and discussed and the research results are supported by citations. This is very well written as well as an interesting and original manuscript. Moreover, this is a very extensive and detailed work. The research results have significant scientific and application value.

Generally, I cannot find any essential errors in the article. I recommend this manuscript for publication after making a few minor revisions.

Detailed conclusions:

 ·         I think the introduction could be extended a little.

·         Page 4, lines 125-126. Please check that the sentence "To this end, 12/20, 8/12, 4/8, 0/4 and filler stockpiles were combined at 20%, 5%, 30%, 40% and 5% by weight of mixture, respectively" is correct? I mean the percentages shown (20%, 5%, 30%, 40% and 5% ). There you can see a double 5% as well as 20% at the beginning.

·         0.25% and 1.5% PE is a very small amount. Please explain why such amounts of PE were used?

·         Is it possible to use larger amounts, e.g. 5 or 10% PE?

·         With such small amounts of PE (0.25% and 1.5%) some changes may be insignificant. Can you relate to it?

·         I think, the authors should also check the literature because there is no formatting consistency.

In my professional opinion no further changes are necessary.

Reviewer 2 Report

Line 110:

“Such PE waste came in the form of a variegated blend of shreds with a density equal to 0.949 Mg/m3” Please clarify what is meant by variegated?

Line 163:

“A pre-conditioning time of at least 4 h was guaranteed to ensure the thermal homogeneity of samples at the selected temperature.” Please comment on thermoreversible aging effects which typically would take longer to reach equilibrium on the order of days and weeks depending on temperature and binder source.

Line 233:

“In the second case, samples were subjected to pulse load repetitions to assess the fracture potential due to dynamic phenomena.” Explain why strain tolerance is not a preferred parameter over numbers of cycles to failure. Can the strain at failure also be reported as that will show a significant reduction for PE-modified mixes.

Line 309:

Figure 6 is for a single binder only. Most Black space diagrams of modified binders are complex. Please make this clear as a limitation of this study and try to provide references to thermorheological complexity in modified asphalt binders and mixes.

Line 310:

Change black to Black.

Lines 354, 409-412:

Increases in stiffness are very significant so will likely lead to premature cracking. Note that pavement trials in Ontario have been disappointing for this reason. Authors may wish to reference recent papers to this effect.

Reviewer 3 Report

Dear Authors,

Please find enclosed my remarks (minor) inserted into submitted by me pdf file. English is correct.

Something which requires your effort is analysis of obtained results in discussion with some from the literature.

Reviewer 4 Report

This is an interesting manuscript. However, the focus was mainly on the engineering side (civil engineering). Thus it may be more suitable for an engineering journal.

A couple comments: (1) The PE wastes used were polyethylene based plastics (with various additives). So this should be stated clearly in the experiment part. (2) Without any specifics, Figure 3 and Figure 4 look like from textbook. The authors should consider improve it. (3) It would be helpful if the authors also show comparisons among REF, PEs02, PEs15 in Figure 8~10 (a couple temp), Figure 14(a)~14(c). 

Reviewer 5 Report

The manuscript illustrates a laboratory characterization study of an asphalt mixture prepared with dry addition of the recycled waste PE. The results provide promising elements towards a successful use of such plastic waste into asphalt mixtures using the dry method. However, the manuscript still needs to be improved. Some comments:

(1) In Section 2, a mixture performance comparison between wet process and dry process should be included. And it is necessary to add an in-depth explanation about the Motivation of this paper.

(2) In Section 3.2, it is suggested to supplement the mixing time for each step.

(3) In L81 you mentioned that 1.5% PE by weight of mixture was chosen to reproduce a typical dry addition allowing to investigate the effect of the plastic content in view of a maximization of waste recycling for environmental reasons. Please explain the reason, which is difficult to understand.

(4) The low-temperature properties of PE-modified asphalt mixtures received a great attention. However, there is no relevant content in this paper. If possible, it is recommended to add relevant content.

Round 2

Reviewer 1 Report

I have no more comments.

Reviewer 2 Report

Authors have addressed the review comments. Hence, paper is acceptable for publication.

Reviewer 3 Report

Dear Authors,

I read your text after correction and found you had followed most of my remarks.

You must have missed something in the line '86' which seems to me gramatically incorrect.

Please check it. For any case I submit the pdf file with your 2nd version.

Reviewer 4 Report

The revised manuscript is acceptable for publication.